# Doubly Calibrated Estimator for Recommendation on Data Missing Not At Random

Anonymous
anonymous@anonymous

## ABSTRACT

Recommender systems often suffer from selection bias as users tend to rate their preferred items. The datasets collected under such conditions exhibit entries missing not at random and thus are not randomized-controlled trials representing the target population. To address this challenge, a doubly robust estimator and its enhanced variants have been proposed as they ensure unbiasedness when accurate imputed errors or predicted propensities are provided. However, we argue that existing estimators rely on miscalibrated imputed errors and propensity scores as they depend on rudimentary models for estimation. We provide theoretical insights into how miscalibrated imputation and propensity models may limit the effectiveness of doubly robust estimators and validate our theorems using real-world datasets. On this basis, we propose a Doubly Calibrated Estimator that involves the calibration of both the imputation and propensity models. To achieve this, we introduce calibration experts that consider different logit distributions across users. Moreover, we devise a tri-level joint learning framework, allowing the simultaneous optimization of calibration experts alongside prediction and imputation models. Through extensive experiments on real-world datasets, we demonstrate the superiority of the Doubly Calibrated Estimator in the context of debiased recommendation tasks.

## CCS CONCEPTS

• **Information systems** → **Collaborative filtering**; **Personalization**; **Recommender systems**.

## KEYWORDS

Recommender System; Collaborative Filtering; Personalization; Recommendation Size; User Utility

**ACM Reference Format:**
Anonymous Author(s). 2024. Doubly Calibrated Estimator for Recommendation on Data Missing Not At Random. In *Proceedings of the ACM Web Conference 2024 (WWW '24), May 13-17, 2024, Singapore.* ACM, New York, NY, USA, 11 pages. https://doi.org/XXXXXXX.XXXXXXX

## 1 INTRODUCTION

Real-world recommender systems utilize feedback data collected from user-item interactions for learning user behaviors. However, there has been recently a growing concern regarding the issue of selection bias in user feedback datasets. As users tend to rate their preferred items, the datasets collected under such conditions exhibit entries missing not at random (MNAR) and thus are not randomized-controlled trials representing the target population. To tackle this issue, numerous debiasing methods have been proposed for unbiased prediction of various user behaviors including explicit ratings [12, 35, 40], implicit feedback [31, 33, 44], and post-click conversion rate [5, 23, 25, 43]. In the early literature, the error-imputation-based (EIB) estimator [36] and the inverse propensity scoring (IPS) estimator [35] are two major approaches for designing unbiased estimators, capable of accurately approximating the ideal loss over the target population with only biased MNAR data.

Lately, the doubly robust (DR) estimator [40] and its enhanced variants [6, 12, 22, 23, 25] are proposed to merge the EIB and the IPS estimator for double robustness. DR estimators ensure the unbiased estimation of the target population when accurate imputed errors or predicted propensities are provided. However, rather than focusing on achieving precise estimation, they prioritize the development of estimators that are robust to the inaccurate estimation of imputed error and propensity score. They enhance the robustness of DR estimators for better bias-variance trade-off by manipulating either the imputed errors [5, 12, 23] or the propensity scores [25]. While DR estimators exhibit state-of-the-art debiasing performance based on their double robustness, we argue that their effectiveness may be limited since they still depend on rudimentary models [13, 19] for estimating imputed errors and propensity scores.

Recent work in the field of machine learning has highlighted that both logistic regression and neural networks, commonly adopted for imputation and propensity models, have a tendency to generate overly confident predictions [1, 11, 20, 21]. The overconfidence problem can be exacerbated especially for DR estimators, as user-item pairs for training the prediction model and the propensity model overlap with each other. DR estimators adopt inverse propensity scoring for the observed pairs that already gave positive signals to the propensity model. Indeed, in our analysis with real-world datasets, we demonstrate that the current imputed error and propensity score estimation yields miscalibrated estimates that fail to accurately reflect the ground-truth likelihood. Miscalibrated imputed errors can increase the bias and the variance of DR estimators and overconfident propensity scores can yield a variance amplification problem. Nonetheless, literature has yet to delve into the direct methodology for training accurate imputation and propensity models, highlighting the necessity for such exploration.

This paper claims that DR estimators can be further enhanced by leveraging model calibration approaches for the imputation and the propensity models. We first provide theoretical insights into how miscalibrated imputation and propensity models may limit the effectiveness of doubly robust estimators: the bias and the variance have an upper bound that is proportional to the calibration errors. On this basis, we propose **Doubly Calibrated Estimator** that involves the calibration of both the imputation and propensity models. To achieve this, we introduce *calibration experts*, each assigned to a specific group of users via the assignment network. By doing so, each expert can learn specialized knowledge about its group for the calibration of imputation and propensity models and also get enough training signals from users within the group. Additionally, we devise a tri-level joint learning framework, allowing the simultaneous optimization of calibration experts alongside prediction and imputation models.

The proposed method offers several merits as follows: (1) Our approach is orthogonal to existing DR estimators and can be seamlessly combined with them, (2) It enables the simultaneous reduction of both bias and variance of DR estimators, (3) It does not require any additional unbiased data. The main contributions of this paper are summarized as follows:

- We provide a theoretical analysis on the correlation between the performance of DR estimators and the calibration of the imputation and the propensity models. Then, we demonstrate existing DR estimators may exhibit limited effectiveness, as they rely on miscalibrated imputed errors and propensity scores.
- We propose Doubly Calibrated Estimator that involves the calibration of both the imputation and propensity models. Imputed errors and propensity scores are calibrated with calibration experts by assigning users to each expert through the assignment network.
- We validate the superiority of the proposed method by extensive experiments on real-world datasets. We also provide in-depth quantitative analyses to verify the effectiveness of each proposed component.

## 2 PRELIMINARIES

### 2.1 Problem Formulation

Let $\mathcal{U} = \{u\}$ and $\mathcal{I} = \{i\}$ denote a set of users and a set of items, respectively. For a pair of $u \in \mathcal{U}$ and $i \in \mathcal{I}$, an observation indicator $o_{u,i}$ is given as 1 if the item $i$ is exposed to the user $u$ and 0 otherwise. The observation indicator $o_{u,i}$ is often referred to as treatment since the rating $r_{u,i}$ is observed only when an item is exposed to a user ($o_{u,i} = 1$) [5, 23, 25]. In this paper, we adopt the binary rating scheme ($r_{u,i} \in \{0, 1\}$) for illustrating purposes, following recent literature in the post-click conversion rate scenario [5, 23, 25, 43] or the implicit feedback scenario [31, 33, 44]. To distinguish the entire population and observed distribution, let $\mathcal{D} = \mathcal{U} \times \mathcal{I}$ denote the set of all user-item pairs and $O = \{(u,i)|(u,i) \in \mathcal{D}, o_{u,i} = 1\}$ denote the set of observed pairs.

If ratings for all user-item pairs in $\mathcal{D}$ are observed, the prediction model $\hat{r}_{u,i} = f_\theta(x_{u,i})$ with input feature $x_{u,i}$ can be trained by the ideal loss function on full observation:

$$\mathcal{L}_{\text{ideal}}(\theta) = \frac{1}{|\mathcal{D}|} \sum_{u,i \in \mathcal{D}} e_{u,i}, \tag{1}$$

where $e_{u,i} = e(\hat{r}_{u,i}, r_{u,i})$ represents the prediction error between the prediction $\hat{r}_{u,i}$ and the target $r_{u,i}$, and can be Mean Squared Error (MSE) [3, 19] or Binary Cross Entropy (BCE) [13]. Instead, the prediction model is often trained by a naive estimator on the only observed ratings, since most ratings are missing due to the users' selection mechanism:

$$\mathcal{E}_{\text{naive}}(\theta) = \frac{1}{|O|} \sum_{u,i \in O} e_{u,i}. \tag{2}$$

For this naive estimator to be an unbiased estimator of the ideal loss, we need the expected value of its estimation across all the possible observations $O$ to precisely match the ideal loss [40], i.e., $\mathbb{E}_O[\mathcal{E}_{\text{naive}}] = \mathcal{L}_{\text{ideal}}$. However, as addressed in [35], the users in the recommender systems tend to rate their preferred items and this process is missing not at random (MNAR). Therefore, the observed set $O$ is not a result of randomized-controlled trials and the naive estimator may induce a large bias [35]: $|\mathbb{E}_O[\mathcal{E}_{\text{naive}}] - \mathcal{L}_{\text{ideal}}| > 0$.

### 2.2 Unbiased Estimators

There have been proposed two major approaches for designing the unbiased estimators. First, Error-imputation-based (EIB) estimators [36, 42] adopt imputation models to directly predict the individual loss $e_{u,i}$ for each pair. Then, EIB estimators use the imputed error as a proxy of errors for missing ratings:

$$\mathcal{E}_{\text{EIB}}(\theta) = \frac{1}{|\mathcal{D}|} \sum_{u,i \in \mathcal{D}} (o_{u,i} e_{u,i} + (1 - o_{u,i})\hat{e}_{u,i}), \tag{3}$$

where $e_{u,i} = e(\hat{r}_{u,i}, r_{u,i})$ is the true error for observed ratings and $\hat{e}_{u,i} = e(\hat{r}_{u,i}, \tilde{r}_{u,i})$ is the imputed error predicted by the imputation model $g_\phi(x_{u,i}) = \tilde{r}_{u,i}$. Obviously, the EIB estimator is an unbiased estimator of the ideal loss when the imputed errors are accurate, i.e, $(\hat{e}_{u,i} = e_{u,i} \ \forall(u,i) \in \mathcal{D}) \Rightarrow (\mathbb{E}_O[\mathcal{E}_{\text{EIB}}] = \mathcal{L}_{\text{ideal}})$.

On the other hand, Inverse Propensity Scoring (IPS) estimators [33, 35] adopt propensity models to predict the probability of observing the true rating. Then, IPS estimators use the propensity score to inversely weight the prediction error for observed ratings:

$$\mathcal{E}_{\text{IPS}}(\theta) = \frac{1}{|\mathcal{D}|} \sum_{u,i \in \mathcal{D}} \frac{o_{u,i} e_{u,i}}{\hat{p}_{u,i}}, \tag{4}$$

where $\hat{p}_{u,i} = \hat{P}(o_{u,i} = 1)$ is the propensity score predicted by the propensity model $h_\psi(x_{u,i}) = \hat{p}_{u,i}$. The IPS estimator is an unbiased estimator of the ideal loss when the predicted propensity scores are accurate, i.e., $(\hat{p}_{u,i} = p_{u,i} \ \forall(u,i) \in O) \Rightarrow (\mathbb{E}_O[\mathcal{E}_{\text{IPS}}] = \mathcal{L}_{\text{ideal}})$.

### 2.3 Doubly Robust Estimators

Recently, Doubly Robust (DR) estimator [40] and its enhanced variants [6, 22, 23, 25] are proposed to merge the EIB and the IPS estimator for double robustness. Given imputed error $\hat{e} = e(\hat{r}, \tilde{r})$ and learned propensity score $\hat{p} = h_\psi(x_{u,i})$, DR estimator can be formulated as follows:

$$\mathcal{E}_{\text{DR}}(\theta) = \frac{1}{|\mathcal{D}|} \sum_{u,i \in \mathcal{D}} \left( \hat{e}_{u,i} + \frac{o_{u,i}(e_{u,i} - \hat{e}_{u,i})}{\hat{p}_{u,i}} \right). \tag{5}$$

By utilizing both the imputed error and the propensity score, DR estimators have double robustness: DR estimator is an unbiased estimator when either the imputed errors or the propensity scores are accurate. The following lemma presents the bias and the variance

of DR estimator induced by the inaccurate estimation of imputed errors and propensity scores.

Lemma 1 (Bias and Variance of DR estimator). *The bias and variance of DR estimator are computed as:*

$$\text{Bias}[\mathcal{E}_{\text{DR}}] = \frac{1}{|\mathcal{D}|} \Big| \sum_{u,i \in \mathcal{D}} \Big( \frac{\hat{p}_{u,i} - p_{u,i}}{\hat{p}_{u,i}} \Big)(e_{u,i} - \hat{e}_{u,i}) \Big|,$$

$$\text{Var}[\mathcal{E}_{\text{DR}}] = \frac{1}{|\mathcal{D}|^2} \sum_{u,i \in \mathcal{D}} \frac{p_{u,i}(1 - p_{u,i})}{\hat{p}_{u,i}^2}(\hat{e}_{u,i} - e_{u,i})^2. \tag{6}$$

Please refer [5, 40] for the proofs.

## 2.4 Limitations of Existing Estimators

According to Lemma 1, the bias and the variance of DR estimator are positively correlated with the inaccuracy of the imputation model and the propensity model. Nevertheless, both models have been learned in a surprisingly simplistic manner. In the early methods [33, 36], heuristic techniques are adopted for the estimation of imputed errors and propensity scores, e.g., $\hat{e}_{u,i} = \omega|\hat{r}_{u,i} - \gamma|$, where $\omega$ and $\gamma$ are hyper-parameters. Recent studies have also embraced straightforward model-based approaches, such as employing logistic regression on held-out unbiased data [25, 35] or training binary classifiers for the observation indicator [5, 23, 40]. These model-based approaches exhibit a slight improvement over heuristic techniques, resulting in a moderately accurate prediction model. Nevertheless, as indicated by recent literature [11, 21], both logistic regression and neural networks have a tendency to generate overly confident predictions. An over-confident propensity model can lead to propensity scores that are either too low or too high, and these poorly calibrated estimates may further hinder the effectiveness of DR estimator.

On the other hand, several recent works [5, 12, 22] focus on developing estimators that are robust to the inaccurate estimation of imputed errors and propensity scores. They enhance the robustness of DR estimator by reducing the bias and the variance while having the same level of inaccuracy in imputed errors and propensity scores. Nevertheless, we argue that their effectiveness may be limited since they still depend on rudimentary models for estimating imputed errors and propensity scores. The literature has yet to delve into the direct methodology for training accurate imputation and propensity models without any unbiased data, highlighting the necessity for such exploration.

## 2.5 Model Calibration

In this paper, we adopt the concept of *model calibration* [11] to quantitatively measure the inaccuracy of the imputation and the propensity models. We say a model is calibrated if its output reflects the ground-truth likelihood of correctness [20]. For the propensity model $h_\psi$ and the observation indicator $o$, a formal definition can be formulated as follows:

$$\mathbb{E}[o|h_\psi(x) = \hat{p}] = \hat{p} \quad \forall \hat{p} \in [0,1]. \tag{7}$$

For example, if we have 100 pairs with propensity scores $\hat{p}_{u,i} = 0.2$, we expect exactly of these pairs to be observed ($o_{u,i} = 1$). Using the above definition, the miscalibration of a propensity model can be measured by Expected Calibration Error (ECE) and Maximum Calibration Error (MCE) [28]:

$$\text{ECE}(h_\psi) = \mathbb{E}_{\hat{p}} \big[ |\mathbb{E}[o|h_\psi(x) = \hat{p}] - \hat{p}| \big],$$

$$\text{MCE}(h_\psi) = \max_{\hat{p}} |\mathbb{E}[o|h_\psi(x) = \hat{p}] - \hat{p}|. \tag{8}$$

ECE and MCE quantify the average and worst-case discrepancy between actual observation proportion and average predicted propensity across $M$ bins. Likewise, we can compute ECE and MCE of the imputation model $g_\phi$ by substituting $(h_\psi, o, \hat{p})$ with $(g_\phi, r, \tilde{r})$ in Eq.8.

## 3 CALIBRATION AND DR ESTIMATORS

In this section, we describe the motivation for proposing our doubly calibrated estimator. We first theoretically analyze how DR estimator stands to benefit from the calibration of the imputation and the propensity models. Then, we empirically demonstrate that existing imputation and propensity models are indeed miscalibrated and limit the effectiveness of DR estimator.

### 3.1 Theoretical Analysis

We analyze how the miscalibration of the imputation and the propensity models amplify the bias and the variance of DR estimator and further hinder the effectiveness of DR estimators.

*3.1.1* **Bias of DR Estimator.** We first present a theorem concerning the interplay between the bias of DR estimator and the calibration of the propensity model.

Theorem 2. *The bias of DR estimator exhibits an upper bound proportional to the calibration error of the propensity model.*

$$\text{Bias}[\mathcal{E}_{\text{DR}}] \le \rho_{\max} \cdot \text{ECE}(h_\psi),$$

$$\text{Bias}[\mathcal{E}_{\text{DR}}] \le \rho_{\max} \cdot \text{MCE}(h_\psi),$$

*where* $\rho_{\max} = \max_{(u,i) \in \mathcal{D}} |(e_{u,i} - \hat{e}_{u,i})/\hat{p}_{u,i}|$.

Please refer to Appendix A for the proof. The above theorem implies that DR estimator may become unreliable when the propensity models are over-confident and yield large calibration errors. Similarly, we can derive the following corollary of Theorem 2 for the imputation model.

Corollary 3. *The bias of DR estimator exhibits an upper bound proportional to the calibration error of the imputation model.*

$$\text{Bias}[\mathcal{E}_{\text{DR}}] \le \pi_{\max} \cdot \text{ECE}(g_\phi),$$

$$\text{Bias}[\mathcal{E}_{\text{DR}}] \le \pi_{\max} \cdot \text{MCE}(g_\phi).$$

*where* $\pi_{\max} = \max_{(u,i) \in \mathcal{D}} |(\hat{p}_{u,i} - p_{u,i})(e_{u,i}^{(1)} - e_{u,i}^{(0)})/\hat{p}_{u,i}|$.

$e^{(r)}$ denotes the loss when the target label is $r$, e.g., $e^{(r)} = -r\log\hat{r} - (1-r)\log(1-\hat{r})$ for BCE. Please refer to Appendix A for the proof. Likewise, the above corollary implies that calibrated imputed errors can reduce the upper bound on the bias of the DR estimator, leading toward the unbiased estimation of the ideal loss.

*3.1.2* **Variance of DR Estimator.** Moreover, we argue that the miscalibration of the imputation model also yields negative effects for the variance of the DR estimator by the following theorem.

THEOREM 4. *The variance of DR estimator exhibits an upper bound proportional to the square of the calibration error of the imputation model.*

$$\text{Var}[\mathcal{E}_{\text{DR}}] \le \omega_{\max} \cdot \left(\text{ECE}(g_\phi)\right)^2,$$

$$\text{Var}[\mathcal{E}_{\text{DR}}] \le \frac{\omega_{\max}}{|\mathcal{D}|} \cdot \left(\text{MCE}(g_\phi)\right)^2,$$

*where* $\omega_{\max} = \max_{(u,i)\in\mathcal{D}} |p_{u,i}(1 - p_{u,i})(e_{u,i}^{(1)} - e_{u,i}^{(0)})^2 / \hat{p}_{u,i}^2|$.

The above theorem implies that the calibration of the imputation model is also able to reduce the upper bound on the variance of the DR estimator. We expect this theorem to give us a new remedy for bias-variance trade-off in the DR estimator as calibrating the pseudo label of the imputation model can reduce both the bias and the variance of the DR estimator simultaneously.

On the other hand, as shown in Lemma 1, the accuracy of the propensity model does not have any relation with the variance of the DR estimator. Nevertheless, the variance is inversely proportional to $\hat{p}_{u,i}^2$, which can be problematic when the estimated propensity scores are exceptionally low. Indeed, current propensity models often produce too low propensity scores as they can easily be over-confident [11, 21], i.e., over-confidence for non-observation. While existing methods [23, 25, 33] incorporate a propensity clipping technique [37] to curtail extremely low propensities, this approach does not address the underlying issue of overconfidence. Instead, we argue that model calibration can effectively mitigate the variance amplification problem, as a calibrated propensity model is less likely to produce extremely low propensities compared to a miscalibrated one.

## 3.2 Empirical Evidence

In this section, we demonstrate that the imputation and propensity models deployed in recent methods are miscalibrated and thus limit the effectiveness of DR estimators, which can be explained by our theoretical analysis. We adopt two DR estimators including DR-JL [40] and TDR [23], and two kinds of imputation and propensity models, including simple heuristics and model-based approaches. (1) For the heuristic approach, we use $\hat{p}_{u,i} = (\sum_{u\in\mathcal{U}} Y_{u,i}/\max_{i\in\mathcal{I}} \sum_{u\in\mathcal{U}} Y_{u,i})^{0.5}$ [33] for the propensity scores and $\hat{e}_{u,i} = \omega|\hat{r}_{u,i} - \gamma|$ [36] for the imputed errors ($\omega$ and $\gamma$ are hyperparameters). (2) For the model-based approach, we adopt the neural collaborative filtering [13] for the imputation and the propensity models. We train the imputation model $g_\phi(x_{u,i}) = \tilde{r}_{u,i}$ by using the imputation loss adopted in [25, 40]:

$$\mathcal{L}_{\text{imp}}(\phi) = \frac{1}{|\mathcal{D}|} \sum_{u,i\in\mathcal{D}} \frac{o_{u,i}(e(\hat{r}, \tilde{r}) - e(\hat{r}, r))^2}{\hat{p}_{u,i}}, \tag{9}$$

and train the propensity model $h_\psi(x_{u,i}) = \hat{p}_{u,i}$ through binary classification between $O$ and $\mathcal{D} \setminus O$, as done in [23].

We investigate the miscalibration of the adopted imputation and propensity models and the performance of DR estimators when they deploy each of the above imputation and propensity models. For the quantitative measurement of the miscalibration, we adopt ECE defined in Eq.8. However, since we cannot observe the true propensity $\mathbb{E}[o|h_\psi(x) = \hat{p}] = P(o = 1|h_\psi(x) = \hat{p})$, we cannot directly compute Eq.8. Instead, we partition the [0,1] range of $\hat{p}$ into $M$ bins and aggregate the value of pairs in each bin as done in

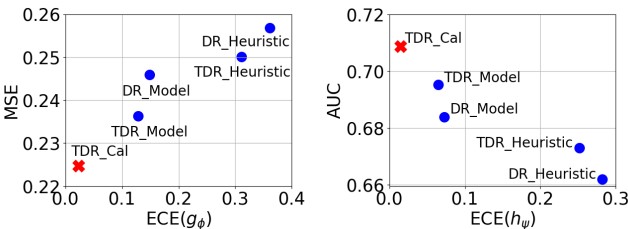

**Figure 1: Performance of DR estimators with various imputation/propensity models on Yahoo!R3 dataset.**

[11, 28]:

$$\text{ECE}_M(h_\psi) = \sum_{m=1}^{M} \frac{|B_m|}{N} \left| \frac{\sum_{(u,i)\in B_m} o_{u,i}}{|B_m|} - \frac{\sum_{(u,i)\in B_m} \hat{p}_{u,i}}{|B_m|} \right| \tag{10}$$

where $B_m$ is $m$-th bin and $N$ is the number of samples. The first term in the absolute value symbols denotes the ground-truth proportion of observations in $B_m$ and the second term denotes the average predicted propensity score of $B_m$. We use $M = 15$ as done in [11, 21].

Figure 1 shows the performance of DR estimators with various imputation and propensity models on Yahoo!R3 dataset[1]. TDR_Cal denotes TDR with our calibrated imputation model and calibrated propensity model (we show this point as a reference). Please note that figures for MSE vs $\text{ECE}(h_\psi)$ and AUC vs $\text{ECE}(g_\phi)$ can be readily reconstructed by Figure 1. We have the following findings: (1) Existing approaches for estimating imputed errors and propensity scores produce poorly calibrated probabilities. This finding is consistent with the previous work [11, 21] concerning that the output of machine learning models, from logistic regression to neural networks, does not necessarily indicate the accurate correctness likelihood. (2) The performance of DR estimators is positively correlated with the calibration of the imputation and the propensity models. We observe that even a naive DR estimator with the model-based approach outperforms TDR with the heuristic approach. This experimental result serves as evidence to support our theoretical analysis in Section 3.1.

## 4 DOUBLY CALIBRATED ESTIMATOR

We propose a **Doubly Calibrated Estimator** on the basis of our comprehensive analysis on the correlation between the performance of DR estimators and the calibration of the imputation and the propensity models. A doubly calibrated estimator entails the calibration of both the imputation and propensity models, achieved through the calibration experts (Section 4.2). These experts are simultaneously optimized alongside prediction and imputation models within our tri-level joint learning framework (Section 4.3).

### 4.1 Motivation

As elaborated upon in Section 3, current imputation and propensity models are miscalibrated, which, as substantiated by our theoretical findings, could potentially constrain the effectiveness of DR estimators. Therefore, DR estimators can benefit from the calibration of imputation and propensity models. The straightforward

---

[1]http://research.yahoo.com/Academic_Relations

approach for the calibration is to adopt a post-processing calibration function $c_\omega : [0,1] \rightarrow [0,1]$ that maps the miscalibrated $\tilde{r}_{u,i} = g_\phi(x_{u,i})$ or $\hat{p}_{u,i} = f_\psi(x_{u,i})$ to the well-calibrated probability. Among various forms of calibration functions, we adopt Platt scaling $c_\omega(p) = \sigma(a \cdot \sigma^{-1}(p) + b)$ [30], a general form of the temperature scaling [11]. Here, $\sigma^{-1}$ is the inverse of the sigmoid function and $\sigma^{-1}(p)$ represents the logit of the probability $p$. Platt scaling has found widespread application across diverse domains, including computer vision [9, 27], natural language processing [7], and recommender system [21].

However, employing a single global calibration function for all users falls short of achieving satisfactory performance. The learning parameters $\omega = \{a, b\}$ represent the characteristics of the logit distribution [20], e.g., $a = \mu_1/\sigma_1^2 - \mu_0/\sigma_0^2$, where $\mu$ and $\sigma$ is the mean and the variance of the logit distribution. As a result, a global calibration function would blend information from users with varying preferences, unable to fully capture the distinct logit distribution of individual users. A naive solution to this challenge is to create a dedicated calibration function for each user: $c_{\omega_u}(p) = \sigma(a_u \cdot \sigma^{-1}(p) + b_u)$, $\forall u \in \mathcal{U}$. Here, the user-specific parameters $\omega_u = \{a_u, b_u\}$ account for the unique logit distributions of user $u$. Nevertheless, this approach also has a limitation. Each calibration function requires training using the respective user's interactions and therefore cold-start users with minimal interactions may not possess sufficient training signals for their calibration function. In the following subsection, we present our solution to strike a balance between the single global calibration and the user-specific calibration.

## 4.2 Calibration Experts

We introduce calibration experts that consider distinct logit distributions across users while alleviating the cold-start problem of user-specific calibration. Inspired by Mixture-of-Experts [15], we deploy $K$ calibration experts, denoted as $\{c_{\omega_k}(p)\}_{k \in [K]}$,[2] with each expert assigned to a specific group of users. By doing so, each expert can learn specialized knowledge about its group for the calibration of imputation and propensity models and also get enough training signals from the users in the group.

Specifically, we devise an assignment network to assign each user to an expert, by mapping the user embedding to the assignment probability:

$$A(E_u) = \alpha_u \in \mathbb{R}^K, \quad (11)$$

where $A : \mathbb{R}^d \rightarrow \mathbb{R}^K$ is the assignment network with a softmax output layer. $E_u \in \mathbb{R}^d$ is the user embedding and $\alpha_u \in \mathbb{R}^K$ is the assignment probability vector of $u$. Each element of the assignment probability vector $\alpha_{u,k}$ represents the probability of the user $u$ being assigned to expert $c_{\omega_k}(p)$. Then, we sample an assignment vector from the assignment probability vector.

$$\beta_u \sim \text{Categorical}(\alpha_u), \quad (12)$$

where $\beta_u$ is a $K$-dimensional one-hot assignment vector sampled from the categorical distribution with the probability $\{\alpha_{u,k}\}_{k \in [K]}$. However, the sampling process is non-differentiable and blocks the gradient flow. To tackle this challenge, we adopt Gumbel-Softmax [16], a continuous relaxation with the reparameterization trick.

---

[2] $[K]$ denotes $\{1, ..., K\}$

With Gumbel-Softmax, the assignment vector $\beta_u$ can be pseudo-sampled as follows:

$$\beta_{u,k} = \frac{\exp((\log\alpha_{u,k} + g_k)/\tau)}{\Sigma_{l=1}^K \exp((\log\alpha_{u,l} + g_l)/\tau)} \quad \forall k \in [K], \quad (13)$$

where $g$ is i.i.d drawn from Gumbel distribution with (location = 0, scale = 1), and $\tau$ is the temperature of the softmax. We use a simple exponential annealing $\tau = T_0(\frac{T_Q}{T_0})^{\frac{q}{Q}}$, where $q$ is the current epoch, $Q$ is the total epochs, $T_0$ is initial temperature and $T_Q$ is the terminal temperature ($T_0 >> T_Q$). With a large $\tau$ the assignment network explores the combinations of the calibration experts, and with a small $\tau$ the assignment network can select a specific calibration expert for a user.

It is noted that we distinguish the set of calibration experts for imputation and propensity models. We represent $\{c_{\omega_k}^{\text{imp}}(p)\}_{k \in [K]}$ as calibration experts for the imputation model and $\{c_{\omega_k}^{\text{prop}}(p)\}_{k \in [K]}$ as calibration experts for the propensity model. Accordingly, the assignment network for the imputation model $A^{\text{imp}}$ takes the embedding of the imputation model $g_\phi$ as an input and the assignment network for the propensity model $A^{\text{prop}}$ takes the embedding of the propensity model $h_\psi$ as an input. Calibration experts and assignment networks are trained in an end-to-end manner as follows: (1) For the imputation model, $\{c_{\omega_k}^{\text{imp}}(p)\}_{k \in [K]}$ and $A^{\text{imp}}$ are trained by the loss

$$\mathcal{L}_{\text{impcal}}(c_{\omega_k}^{\text{imp}}, A^{\text{imp}}) = \sum_{(u,i) \in O_{\text{val}}} -\frac{r_{u,i}}{\bar{p}_{u,i}}\log\left(\sum_{k=1}^K \beta_{u,k}^{\text{imp}} \cdot c_{\omega_k}^{\text{imp}}(\tilde{r})\right)$$
$$- \left(1 - \frac{r_{u,i}}{\bar{p}_{u,i}}\right)\log\left(1 - \sum_{k=1}^K \beta_{u,k}^{\text{imp}} \cdot c_{\omega_k}^{\text{imp}}(\tilde{r})\right), \quad (14)$$

where $\tilde{r}_{u,i} = g_\phi(x_{u,i})$ is the miscalibrated pseudo label of the imputation model and $\bar{p}_{u,i} = \sum_{k=1}^K \beta_{u,k}^{\text{prop}} \cdot c_{\omega_k}^{\text{prop}}(\hat{p})$ is the calibrated propensity score. $O_{\text{val}}$ is the set of observed pairs in the validation set. It is a common practice to adopt validation set as the calibration set [9, 11, 20, 21]. We use the binary cross-entropy loss, a widely adopted loss function for the calibration of binary classifiers [11, 20, 21]. (2) For the propensity model, similarly, $\{c_{\omega_k}^{\text{prop}}(p)\}_{k \in [K]}$ and $A^{\text{prop}}$ are trained by the loss

$$\mathcal{L}_{\text{propcal}}(c_{\omega_k}^{\text{prop}}, A^{\text{prop}}) = \sum_{(u,i,o) \in \mathcal{D}_{\text{val}}} -o_{u,i}\log\left(\sum_{k=1}^K \beta_{u,k}^{\text{prop}} \cdot c_{\omega_k}^{\text{prop}}(\hat{p})\right)$$
$$- (1 - o_{u,i})\log\left(1 - \sum_{k=1}^K \beta_{u,k}^{\text{prop}} \cdot c_{\omega_k}^{\text{prop}}(\hat{p})\right), \quad (15)$$

where $\hat{p}_{u,i} = h_\psi(x_{u,i})$ is the miscalibrated propensity score and $\mathcal{D}_{\text{val}} = \{(u, i, o_{u,i} = 1)|(u, i) \in O_{\text{val}}\} \cup \{(u, i, o_{u,i} = 0)|(u, i) \in \mathcal{D} \setminus (O_{\text{val}} \cup O)\}$. During the training of the calibration experts, the imputation model $g_\phi(x_{u,i}) = \tilde{r}_{u,i}$ and the propensity model $h_\psi(x_{u,i}) = \hat{p}_{u,i}$ are frozen.

## 4.3 Tri-Level Joint Learning Framework

We introduce a tri-level joint learning framework where the proposed calibration experts can be simultaneously optimized along

---

**Algorithm 1:** Tri-level Joint Learning Framework

---

**Input** : $\mathcal{D}, O, O_{\text{val}}$, pre-trained propensity model $f_\psi$,
pre-trained calibration experts $\{c_{\omega_k}^{\text{prop}}(p)\}_{k \in [K]}$,
pre-trained assignment network $A^{\text{prop}}$.

1 **while** *epochs remain or stopping criteria is not satisfied* **do**

2    **for** *mini-batch $O^s$ of $O$* **do**

3       Update imputation model $g_\phi$ with $\mathcal{L}_{imp}^{\text{cal}}$ (Eq.16)

4       Sample a batch of user-item pairs from $\mathcal{D}$

5       Update prediction model $f_\theta$ with $\mathcal{E}_{DR}^{\text{cal}}$ (Eq.17)

6    **for** *mini-batch $O_{\text{val}}^s$ of $O_{\text{val}}$* **do**

7       Update calibration experts $\{c_{\omega_k}^{\text{imp}}(p)\}_{k \in [K]}$ and
assignment network $A^{\text{imp}}$ with $\mathcal{L}_{\text{impcal}}$ (Eq.14)

---

with the existing DR estimators. We have four core components including prediction model $f_\theta(x_{u,i}) = \hat{r}_{u,i}$, imputation model $g_\phi(x_{u,i}) = \tilde{r}_{u,i}$, propensity model $h_\psi(x_{u,i}) = \hat{p}_{u,i}$, and calibration experts $\{c_{\omega_k}(p)\}_{k \in [K]}$ for the proposed doubly calibrated estimator. Among them, the propensity model can be trained by itself as the other three components are not related to its optimization process. In this paper, we train the propensity model through binary classification between $O$ and $\mathcal{D} \setminus O$, as done in Section 3.2. Then, on top of the frozen propensity model, we train the calibration experts $\{c_{\omega_k}^{\text{prop}}(p)\}_{k \in [K]}$ and the assignment network $A^{\text{prop}}$ with $\mathcal{L}_{\text{propcal}}$ in Eq.15.

The other components need to be trained together since their loss functions incorporate predictions from others. In our tri-level joint learning framework, each component is trained as follows: (1) The imputation model $g_\phi(x_{u,i}) = \tilde{r}_{u,i}$ is optimized by using the imputation loss function with the calibrated propensity scores. For illustrative purposes, we show the imputation loss adopted in [25, 40] with the calibrated propensity scores:

$$\mathcal{L}_{\text{imp}}^{\text{cal}}(\phi) = \frac{1}{|\mathcal{D}|} \sum_{u,i \in \mathcal{D}} \frac{o_{u,i}(e(\hat{r}, \tilde{r}) - e(\hat{r}, r))^2}{\bar{p}_{u,i}}, \qquad (16)$$

where $\bar{p}_{u,i} = \sum_{k=1}^{K} \beta_{u,k}^{\text{prop}} \cdot c_{\omega_k}^{\text{prop}}(\hat{p})$ is the calibrated propensity score. (2) The calibration experts $\{c_{\omega_k}^{\text{imp}}(p)\}_{k \in [K]}$ and the assignment network $A^{\text{imp}}$ are trained with $\mathcal{L}_{\text{impcal}}$ in Eq.14 on top of the frozen imputation model. (3) Lastly, the prediction model $f_\theta$ is trained with the calibrated imputed errors and the calibrated propensity scores:

$$\mathcal{E}_{\text{DR}}^{\text{cal}}(\theta) = \frac{1}{|\mathcal{D}|} \sum_{u,i \in \mathcal{D}} \left( \bar{e}_{u,i} + \frac{o_{u,i}(e_{u,i} - \bar{e}_{u,i})}{\bar{p}_{u,i}} \right), \qquad (17)$$

where $\bar{p}_{u,i}$ is the calibrated propensity score and $\bar{e}_{u,i} = e(\hat{r}, \bar{r})$ is the calibrated imputed error with calibrated pseudo label $\bar{r}_{u,i} = \sum_{k=1}^{K} \beta_{u,k}^{\text{imp}} \cdot c_{\omega_k}^{\text{imp}}(\tilde{r})$. Algorithm 1 presents the entire procedure of the tri-level joint learning framework. It is noted that the proposed method is orthogonal to existing DR estimators and can be used in tandem with them. We can adopt any other existing imputation loss [5, 12, 23] instead of Eq.16, by substituting the miscalibrated propensity score $\hat{p}$ with our calibrated propensity score $\bar{p}$.

**Table 1: Data statistics.**

| Dataset | #Users | #Items | #Training data | #Test data |
|---|---|---|---|---|
| Coat | 290 | 300 | 6,960 | 4,640 |
| Yahoo!R3 | 15,401 | 1,001 | 311,704 | 54,000 |
| KuaiRec | 7,163 | 10,596 | 201,171 | 117,113 |

## 5 EXPERIMENTS

### 5.1 Experiment Setup

*5.1.1 Datasets.* To evaluate the performance of unbiased recommendations, we need an unbiased test set along with an MNAR training set. Following recent studies [5, 23, 35, 40], we adopt two real-world datasets, including Coat [35] and Yahoo!R3[3]. Additionally, we adopt KuaiRec [10], a recently published dataset that has a separate unbiased test set where the users are asked to rate all test items. For Coat and Yahoo!R3, the observed rating $r_{u,i}$ is set to 1 if the explicit rating is greater than 3 and is set to 0 otherwise, as done in compared studies [5, 23, 25, 31]. For KuaiRec, the rating $r_{u,i}$ is set to 1 if the watching ratio is greater than 1 and is set to 0 otherwise. Data statistics are presented in Table 1. We hold out 10% of the training set as the validation set.

*5.1.2 Compared methods.* We validate the superiority of Doubly Calibrated Estimator with various debiasing methods as follows: (1) IPS estimators: IPS [35], SNIPS [38], AT [31], (2) EIB estimator: CVIB [42], (3) Multi-task approaches: Multi-DR [43], ESCM$^2$-DR [39], (4) DR estimators: DR-JL [40], MRDR-JL [12], BRD-DR [6], MR [22], DR-MSE [5], StableDR [25], TDR-CL [23]. Please note that we exclude methods utilizing a fraction of the unbiased test set in the training phase [2, 4, 8, 24, 41] for a fair comparison.

*5.1.3 Implementation details.*
**Compared methods.** For all methods compared, we use their author codes and strictly follow the parameters and configurations as documented in their papers and codes. For methods without any public source code, we use PyTorch [29] for the implementation. Additionally, in cases where the best configurations are not provided for each dataset, we perform a grid search for hyperparameters on the validation set. Following previous studies [23, 25, 31, 33, 35, 40], the base architectures of the prediction, imputation, and propensity models are chosen from either matrix factorization [19] or neural collaborative filtering [13], based on the validation performance. It is noted that we do not use the unbiased test set for the propensity estimation for a fair comparison.
**Proposed method.** For our Doubly Calibrated Estimator (DCE), we devise two variants: **DCE-DR** utilizing $\mathcal{L}_{\text{imp}}^{\text{cal}}$ in Eq.16 and **DCE-TDR** utilizing the imputation loss of TDR [23] instead of Eq.16 with substituting the miscalibrated propensity score $\hat{p}$ with our calibrated propensity score $\bar{p}$. For the assignment network, we use a single-layer MLP with the softmax output layer. For the Gumbel-Softmax, the initial temperature is set to 1 and the terminal temperature is set to $10^{-3}$. We set the batch size as 128 for Coat, 4096 for Yahoo!R3 and KuaiRec. The other hyperparameters are tuned by using grid searches on the validation set. We use Adam optimizer [18] where the learning rate and weight decay are chosen from the range $[10^{-4}, 10^{-1}]$. The number of experts $K$ is chosen from

---

[3]http://research.yahoo.com/Academic_Relations

**Table 2: Recommendation performance. The numbers in bold indicate the best results, while those underlined represent the best competitor. * denotes the statistical significance of the paired t-test at a 0.05 level when compared to the best competitor.**

| Method | Coat | | | | Yahoo!R3 | | | | KuaiRec | | | |
|---|---|---|---|---|---|---|---|---|---|---|---|---|
| | MSE | AUC | N@5 | N@10 | MSE | AUC | N@5 | N@10 | MSE | AUC | N@50 | N@100 |
| Naive | 0.2547 | 0.6828 | 0.6087 | 0.6747 | 0.2603 | 0.6528 | 0.6303 | 0.7584 | 0.2738 | 0.7510 | 0.7350 | 0.7571 |
| SNIPS | 0.2438 | 0.7061 | 0.6145 | 0.6875 | 0.2493 | 0.6815 | 0.6451 | 0.7701 | 0.2673 | 0.7608 | 0.7507 | 0.7844 |
| IPS | 0.2423 | 0.6935 | 0.6130 | 0.6824 | 0.2496 | 0.6757 | 0.6348 | 0.7641 | 0.2699 | 0.7601 | 0.7442 | 0.7768 |
| CVIB | 0.2311 | 0.7006 | 0.6221 | 0.6991 | 0.2438 | 0.6823 | 0.6487 | 0.7709 | 0.2702 | 0.7540 | 0.7381 | 0.7612 |
| AT | 0.2332 | 0.7020 | 0.6302 | 0.6984 | 0.2480 | 0.6816 | 0.6419 | 0.7667 | 0.2631 | 0.7581 | 0.7471 | 0.7803 |
| DR-JL | 0.2302 | 0.7122 | 0.6382 | 0.7042 | 0.2459 | 0.6837 | 0.6515 | 0.7731 | 0.2580 | 0.7683 | 0.7689 | 0.7932 |
| | ± 0.0013 | ± 0.0053 | ± 0.0088 | ± 0.0054 | ± 0.0016 | ± 0.0025 | ± 0.0036 | ± 0.0020 | ± 0.0018 | ± 0.0037 | ± 0.0034 | ± 0.0035 |
| Multi-DR | 0.2288 | 0.7201 | 0.6410 | 0.7081 | 0.2407 | 0.6863 | 0.6587 | 0.7712 | 0.2557 | 0.7714 | 0.7771 | 0.7985 |
| MRDR-JL | 0.2193 | 0.7192 | 0.6360 | 0.7016 | 0.2394 | 0.6842 | 0.6602 | 0.7727 | 0.2542 | 0.7705 | 0.7825 | 0.8023 |
| ESCM$^2$-DR | 0.2201 | 0.7256 | 0.6353 | 0.7022 | 0.2401 | 0.6932 | 0.6683 | 0.7693 | 0.2520 | 0.7811 | 0.7921 | 0.8049 |
| BRD-DR | 0.2196 | 0.7286 | 0.6441 | 0.7094 | 0.2382 | 0.6850 | 0.6592 | 0.7708 | 0.2505 | 0.7831 | 0.7945 | 0.8051 |
| MR | 0.2181 | 0.7243 | 0.6436 | 0.7118 | 0.2312 | 0.6923 | 0.6691 | 0.7747 | 0.2496 | 0.7851 | 0.7912 | 0.8017 |
| DR-MSE | 0.2152 | 0.7214 | 0.6417 | 0.7089 | 0.2377 | 0.6872 | 0.6660 | 0.7724 | 0.2510 | 0.7792 | 0.7823 | 0.8067 |
| | ± 0.0018 | ± 0.0042 | ± 0.0066 | ± 0.0064 | ± 0.0012 | ± 0.0018 | ± 0.0022 | ± 0.0011 | ± 0.0011 | ± 0.0018 | ± 0.0027 | ± 0.0032 |
| StableDR | 0.2238 | 0.7167 | 0.6335 | 0.7073 | 0.2425 | 0.6891 | 0.6613 | 0.7699 | 0.2543 | 0.7714 | 0.7863 | 0.7976 |
| | ± 0.0028 | ± 0.0061 | ± 0.0087 | ± 0.0086 | ± 0.0033 | ± 0.0080 | ± 0.0048 | ± 0.0029 | ± 0.0020 | ± 0.0061 | ± 0.0052 | ± 0.0048 |
| TDR-CL | 0.2173 | 0.7302 | 0.6425 | 0.7121 | 0.2363 | 0.6951 | 0.6708 | 0.7763 | 0.2477 | 0.7885 | 0.7907 | 0.8037 |
| | ± 0.0005 | ± 0.0045 | ± 0.0098 | ± 0.0050 | ± 0.0010 | ± 0.0015 | ± 0.0010 | ± 0.0013 | ± 0.0006 | ± 0.0008 | ± 0.0011 | ± 0.0012 |
| DCE-DR | 0.2109 | 0.7384 | 0.6581 | 0.7232 | 0.2281 | 0.7041 | 0.6826 | 0.7923 | 0.2430 | 0.8058 | 0.8079 | 0.8122 |
| | ± 0.0004 | ± 0.0019 | ± 0.0030 | ± 0.0043 | ± 0.0027 | ± 0.0013 | ± 0.0018 | ± 0.0014 | ± 0.0005 | ± 0.0006 | ± 0.0022 | ± 0.0016 |
| DCE-TDR | **0.2054*** | **0.7412*** | **0.6623*** | **0.7258*** | **0.2247*** | **0.7087*** | **0.6901*** | **0.8014*** | **0.2394*** | **0.8097*** | **0.8162*** | **0.8249*** |
| | ± 0.0006 | ± 0.0032 | ± 0.0045 | ± 0.0061 | ± 0.0014 | ± 0.0015 | ± 0.0016 | ± 0.0012 | ± 0.0010 | ± 0.0012 | ± 0.0021 | ± 0.0026 |

{5, 10, 20} and the embedding size is chosen from {16, 32, 64}. We will provide the GitHub repository of the proposed method in the final version.

## 5.2 Performance Comparison

Table 2 shows the recommendation performance of the proposed method and the compared debiasing methods. We use MSE [3], AUC [14], NDCG@$K$ (N@$K$) [17] for the evaluation metrics, following the conventions in the recent studies [5, 23, 25]. Due to the lack of space, we report standard deviation only for DR-JL and its recent variants. We report the average result of five independent runs. We first observe that both the IPS estimators and the EIB estimator outperform the Naive estimator, indicating the importance of addressing the MNAR problem in recommendation datasets. Also, DR-JL and its variants perform better than the IPS and the EIB estimator based on their double robustness. The state-of-the-art DR estimators manipulate the imputed errors [5, 23] or the propensity scores [25] for the better bias-variance trade-off and show enhanced unbiased recommendation performance. However, their effectiveness has been limited as they still utilize the miscalibrated imputed errors and propensity scores. The proposed methods (DCE-DR and DCE-TDR) employ calibration experts for both the imputation and the propensity models, effectively generating calibrated estimates which are critical requirements for ensuring the unbiasedness of DR estimators. The calibration experts, the prediction model, and the imputation model are optimized simultaneously in our tri-level

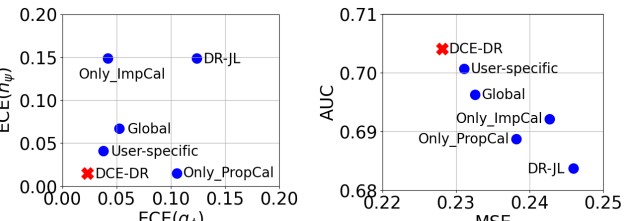

**Figure 2: Ablation study on Yahoo!R3.**

joint learning framework, and successfully increase the recommendation performance from the existing DR estimators by up to 8.37% (DCE-DR vs DR-JL in MSE on Coat).

## 5.3 Ablation Study

Figure 2 shows the ablation study of the proposed doubly calibrated estimator. We introduce four ablated methods crafted to showcase the superiority of our design choices: (1) 'Global' indicates DCE-DR with global calibration function $c_\omega$ for all users. (2) 'User-specific' represents DCE-DR with user-specific calibration functions $\{c_{\omega_u}\}_{u \in \mathcal{U}}$. (3) 'Only_ImpCal' denotes DCE-DR with calibration experts only for the imputation model. (4) 'Only_PropCal' denotes DCE-DR with calibration experts only for the propensity model.

First, we observe that our calibration experts outperform the global and the user-specific calibration (i.e., DCE-DR vs Global, User-specific). Calibration experts can consider distinct logit distributions and learn specialized information about their group. Also,

**Table 3: Time and space analysis. Time denotes the wall time (in s) used for the training and #params. denotes the number of learnable parameters. Please note that the base model and the embedding size are chosen with the grid search on the validation set.**

| Method | Coat | | Yahoo!R3 | | KuaiRec | |
|--------|------|------|----------|------|---------|------|
| | Time | #params. | Time | #params. | Time | #params. |
| MF | 8.16 | 9,440 | 98.87 | 524,864 | 211.47 | 1,136,576 |
| IPS | 12.14 | 18,913 | 217.32 | 1,049,793 | 481.25 | 2,273,281 |
| DR-JL | 19.04 | 28,353 | 460.04 | 3,149,313 | 538.39 | 3,409,857 |
| DR-MSE | 22.14 | 37,772 | 308.01 | 2,099,488 | 697.18 | 4,546,432 |
| TDR-CL | 24.58 | 28,354 | 319.86 | 1,574,660 | 702.74 | 3,409,864 |
| DCE-DR | 24.94 | 28,523 | 320.52 | 1,575,317 | 710.91 | 3,412,457 |
| DCE-TDR | 26.21 | 28,524 | 346.43 | 1,575,532 | 752.13 | 3,412,464 |

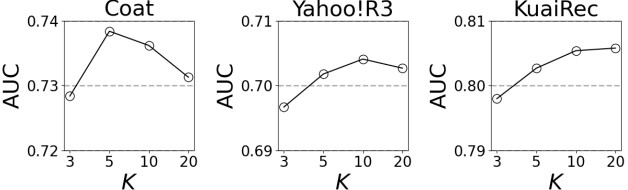

**Figure 3: Hyper-parameter study on number of experts ($K$) with DCE-DR.**

they alleviate the cold-start problem of user-specific calibration by gathering enough training signals from the users in the group. As a result, the calibration experts produce more calibrated imputed errors and propensity scores (Figure 2 left), and accordingly, DCE-DR with calibration experts outperforms DCE-DR with global or user-specific calibration in terms of the unbiased recommendation performance (Figure 2 right).

Second, we observe that employing calibration experts for both the imputation and the propensity model is important for the effectiveness of DCE-DR (i.e., DCE-DR vs Only_ImpCal, Only_PropCal). As theoretically demonstrated in Section 3.1, the bias and the variance of DR estimators are positively correlated with the miscalibration of both the imputation and the propensity models. Moreover, the calibrated propensity scores enhance the calibration (Eq.14) and the optimization (Eq.16) of the imputation model. Therefore, DCE-DR exhibits superior performance when utilizing both calibrated imputed errors and calibrated propensity scores, as compared to employing either one individually.

### 5.4 Time and Space Analysis

Table 3 shows the training time and the number of learnable parameters of compared methods. In this experiment, We use GTX Titan Xp GPU and Intel Xeon(R) E5-2667 v4 CPU. First, we observe that the proposed doubly calibrated estimators do not increase the training time significantly (less than 10% compared to TDR-CL). Each calibration expert is equivalent to the logistic regression and can be efficiently solved. Also, the calibrated experts are updated within 10-20 iterations on the small validation set. Second, doubly calibrated estimators introduce only marginal additional learning parameters in comparison to existing methods. Each calibration expert has two parameters and thus the total number of additional parameters is

$(4 + 2d)K$ ($4K$ for calibration experts and $2dK$ for the assignment networks). Here, $d \leq 64$ and $K \leq 20$ (the hyper-parameter study on $K$ is presented in Figure 3).

## 6 RELATED WORK

Over the past recent years, debiasing methods have been extremely investigated for accurately approximating the ideal loss over the entire population with only biased MNAR data. EIB estimator [36, 42] and IPS estimator [33, 35] are two major approaches for designing unbiased estimators. Lately, DR estimator [40] and its enhanced variants [6, 12, 22, 23, 25, 39, 43] are proposed to merge EIB and IPS estimators for double robustness, and have shown state-of-the-art performance for various user behaviors including explicit ratings [12, 35, 40], implicit feedback [31, 33, 44], uplift modeling [32, 34], and post-click conversion rate [5, 23, 25, 43]. The recent trend is focusing on the development of estimators that are robust to the inaccurate estimation of imputed error and propensity score, rather than achieving precise estimation of them. Recent studies enhance the robustness of DR estimators by manipulating the imputed errors [5, 12, 23] and the propensity scores [25], or by adopting multiple imputation and propensity models [22]. Nonetheless, their effectiveness may be limited since they still depend on miscalibrated imputed errors and propensity scores.

On the other hand, some debiasing methods [2, 4, 8, 24, 26, 41] utilize a small set of unbiased data during the training phase to enhance debiasing performance. For instance, approaches like [25, 33, 35] employ logistic regression on held-out unbiased data for accurate propensity estimation. Other studies leverage unbiased data to address unobserved confounding in selection bias [24, 41] and for causal inference [2, 4]. These methods exhibit a slight improvement compared to scenarios without unbiased uniform data, resulting in a moderately accurate prediction model. Nonetheless, obtaining unbiased data in real-world applications can be challenging, as employing a random exposure policy may degrade user satisfaction. Indeed, they utilize a small fraction of an unbiased test set for utilizing held-out uniform data. In contrast, our doubly calibrated estimator operates without any unbiased data and effectively estimates precise imputed errors and propensity scores.

## 7 CONCLUSION

We demonstrate that current estimators rely on miscalibrated imputed errors and propensity scores and provide theoretical analysis that highlights the potential limitations of doubly robust estimators when faced with miscalibration in imputation and propensity models. Building upon these findings, we propose a Doubly Calibrated Estimator, which involves calibrating both the imputation and propensity models. To achieve this, we introduce calibration experts that consider different logit distributions across users. Furthermore, we design a tri-level joint learning framework, enabling the simultaneous optimization of calibration experts alongside prediction and imputation models. Through extensive experiments conducted on real-world datasets, we demonstrate the enhanced performance of the doubly calibrated estimator in debiased recommendation tasks. We anticipate that our doubly calibrated estimator can be seamlessly integrated with various DR estimators in future work.

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

# Appendix

## A  PROOFS

THEOREM 2. *The bias of DR estimator exhibits an upper bound proportional to the calibration error of the propensity model.*

$$\text{Bias}[\mathcal{E}_{\text{DR}}] \le \rho_{\max} \cdot \text{ECE}(h_\psi),$$
$$\text{Bias}[\mathcal{E}_{\text{DR}}] \le \rho_{\max} \cdot \text{MCE}(h_\psi),$$

*where* $\rho_{\max} = \max_{(u,i) \in \mathcal{D}} |(e_{u,i} - \hat{e}_{u,i})/\hat{p}_{u,i}|$.

PROOF.

(1)

$$\text{Bias}[\mathcal{E}_{\text{DR}}] = \frac{1}{|\mathcal{D}|} \Big| \sum_{u,i \in \mathcal{D}} \Big( \frac{\hat{p}_{u,i} - p_{u,i}}{\hat{p}_{u,i}} \Big) (e_{u,i} - \hat{e}_{u,i}) \Big|$$

$$\le \frac{1}{|\mathcal{D}|} \sum_{u,i \in \mathcal{D}} |\hat{p}_{u,i} - p_{u,i}| \Big| \frac{e_{u,i} - \hat{e}_{u,i}}{\hat{p}_{u,i}} \Big|$$

$$\le \rho_{\max} \cdot \frac{1}{|\mathcal{D}|} \sum_{u,i \in \mathcal{D}} |\hat{p}_{u,i} - p_{u,i}|,$$

where $\rho_{\max} = \max_{(u,i) \in \mathcal{D}} |(e_{u,i} - \hat{e}_{u,i})/\hat{p}_{u,i}|$. From Eq.8, we get the empirical ECE over $\mathcal{D}$.

$$\text{ECE}(h_\psi) = \mathbb{E}_{\hat{p}} \big[ |\mathbb{E}[o|h_\psi(x) = \hat{p}] - \hat{p}| \big]$$

$$= \sum_{\hat{p}} |P(o = 1|h_\psi(x) = \hat{p}) - \hat{p}| \cdot P(\hat{p})$$

$$= \frac{1}{|\mathcal{D}|} \sum_{u,i \in \mathcal{D}} |P(o_{u,i} = 1) - \hat{p}_{u,i}|$$

$$= \frac{1}{|\mathcal{D}|} \sum_{u,i \in \mathcal{D}} |p_{u,i} - \hat{p}_{u,i}|$$

By combining the above two equations, we get

$$\text{Bias}[\mathcal{E}_{\text{DR}}] \le \rho_{\max} \cdot \text{ECE}(h_\psi).$$

(2)
Similar to (1), we get

$$\text{Bias}[\mathcal{E}_{\text{DR}}] \le \rho_{\max} \cdot \frac{1}{|\mathcal{D}|} \sum_{u,i \in \mathcal{D}} |\hat{p}_{u,i} - p_{u,i}|$$

$$\le \rho_{\max} \cdot \max_{u,i \in \mathcal{D}} |\hat{p}_{u,i} - p_{u,i}|$$

$$\text{MCE}(h_\psi) = \max_{\hat{p} \in [0,1]} |\mathbb{E}[o|h_\psi(x) = \hat{p}] - \hat{p}|$$

$$= \max_{\hat{p} \in [0,1]} |P(o = 1|h_\psi(x) = \hat{p}) - \hat{p}|$$

$$= \max_{u,i \in \mathcal{D}} |P(o_{u,i} = 1) - \hat{p}_{u,i}|$$

$$= \max_{u,i \in \mathcal{D}} |p_{u,i} - \hat{p}_{u,i}|.$$

Combining the above two equations, we get

$$\text{Bias}[\mathcal{E}_{\text{DR}}] \le \rho_{\max} \cdot \text{MCE}(h_\psi).$$

$\square$

COROLLARY 3. *The bias of DR estimator exhibits an upper bound proportional to the calibration error of the imputation model.*

$$\text{Bias}[\mathcal{E}_{\text{DR}}] \le \pi_{\max} \cdot \text{ECE}(g_\phi),$$
$$\text{Bias}[\mathcal{E}_{\text{DR}}] \le \pi_{\max} \cdot \text{MCE}(g_\phi).$$

*where* $\pi_{\max} = \max_{(u,i) \in \mathcal{D}} |(\hat{p}_{u,i} - p_{u,i})(e_{u,i}^{(1)} - e_{u,i}^{(0)})/\hat{p}_{u,i}|$.

PROOF.
(1)

$$\text{Bias}[\mathcal{E}_{\text{DR}}] = \frac{1}{|\mathcal{D}|} \Big| \sum_{u,i \in \mathcal{D}} \Big( \frac{\hat{p}_{u,i} - p_{u,i}}{\hat{p}_{u,i}} \Big) (e_{u,i} - \hat{e}_{u,i}) \Big|$$

$$\le \frac{1}{|\mathcal{D}|} \sum_{u,i \in \mathcal{D}} \Big| \frac{\hat{p}_{u,i} - p_{u,i}}{\hat{p}_{u,i}} \Big| |e_{u,i} - \hat{e}_{u,i}|$$

$$= \frac{1}{|\mathcal{D}|} \sum_{u,i \in \mathcal{D}} \Big| \frac{\hat{p}_{u,i} - p_{u,i}}{\hat{p}_{u,i}} \Big| |P(r_{u,i} = 1) - \tilde{r}_{u,i}| |e_{u,i}^{(1)} - e_{u,i}^{(0)}|$$

$$\because e - \hat{e} = P(r = 1)e^{(1)} + (1 - P(r = 1))e^{(0)} - \tilde{r}e^{(1)} - (1 - \tilde{r})e^{(0)}$$

$$= (P(r = 1) - \tilde{r})(e^{(1)} - e^{(0)})$$

With $\pi_{\max} = \max_{(u,i) \in \mathcal{D}} |(\hat{p}_{u,i} - p_{u,i})(e_{u,i}^{(1)} - e_{u,i}^{(0)})/\hat{p}_{u,i}|$, we get

$$\text{Bias}[\mathcal{E}_{\text{DR}}] \le \pi_{\max} \cdot \frac{1}{|\mathcal{D}|} \sum_{u,i \in \mathcal{D}} |P(r_{u,i} = 1) - \tilde{r}_{u,i}|.$$

Also, similar to the proof of Theorem 2, we get

$$\text{ECE}(g_\phi) = \frac{1}{|\mathcal{D}|} \sum_{u,i \in \mathcal{D}} |P(r_{u,i} = 1) - \tilde{r}_{u,i}|.$$

By combining the above two equations, we get

$$\text{Bias}[\mathcal{E}_{\text{DR}}] \le \pi_{\max} \cdot \text{ECE}(g_\phi).$$

(2)
Similar to (1), we get

$$\text{Bias}[\mathcal{E}_{\text{DR}}] \le \pi_{\max} \cdot \frac{1}{|\mathcal{D}|} \sum_{u,i \in \mathcal{D}} |P(r_{u,i} = 1) - \tilde{r}_{u,i}|$$

$$\le \pi_{\max} \cdot \max_{u,i \in \mathcal{D}} |P(r_{u,i} = 1) - \tilde{r}_{u,i}|$$

$$\text{MCE}(g_\phi) = \max_{\tilde{r} \in [0,1]} |\mathbb{E}[r|g_\phi(x) = \tilde{r}] - \tilde{r}|$$

$$= \max_{\tilde{r} \in [0,1]} |P(r = 1|g_\phi(x) = \tilde{r}) - \tilde{r}|$$

$$= \max_{u,i \in \mathcal{D}} |P(r_{u,i} = 1) - \tilde{r}_{u,i}|.$$

By combining the above two equations, we get

$$\text{Bias}[\mathcal{E}_{\text{DR}}] \le \pi_{\max} \cdot \text{MCE}(g_\phi).$$

$\square$

THEOREM 4. *The variance of DR estimator exhibits an upper bound proportional to the square of the calibration error of the imputation model.*

$$\text{Var}[\mathcal{E}_{\text{DR}}] \leq \omega_{\max} \cdot \big(\text{ECE}(g_\phi)\big)^2,$$

$$\text{Var}[\mathcal{E}_{\text{DR}}] \leq \frac{\omega_{\max}}{|\mathcal{D}|} \cdot \big(\text{MCE}(g_\phi)\big)^2,$$

*where* $\omega_{\max} = \max_{(u,i) \in \mathcal{D}} |p_{u,i}(1 - p_{u,i})(e_{u,i}^{(1)} - e_{u,i}^{(0)})^2 / \hat{p}_{u,i}^2|$.

PROOF.

(1)

$$
\begin{aligned}
\text{Var}[\mathcal{E}_{\text{DR}}] &= \frac{1}{|\mathcal{D}|^2} \sum_{u,i \in \mathcal{D}} \frac{p_{u,i}(1 - p_{u,i})}{\hat{p}_{u,i}^2} (\hat{e}_{u,i} - e_{u,i})^2 \\
&= \frac{1}{|\mathcal{D}|^2} \sum_{u,i \in \mathcal{D}} \left| \frac{p_{u,i}(1 - p_{u,i})}{\hat{p}_{u,i}^2} \right| |P(r_{u,i} = 1) - \tilde{r}_{u,i}|^2 |e_{u,i}^{(1)} - e_{u,i}^{(0)}|^2 \\
&\leq \omega_{\max} \cdot \frac{1}{|\mathcal{D}|^2} \sum_{u,i \in \mathcal{D}} |P(r_{u,i} = 1) - \tilde{r}_{u,i}|^2 \\
&\leq \omega_{\max} \cdot \frac{1}{|\mathcal{D}|^2} \left( \sum_{u,i \in \mathcal{D}} |P(r_{u,i} = 1) - \tilde{r}_{u,i}| \right)^2 \\
&= \omega_{\max} \cdot \big(\text{ECE}(g_\phi)\big)^2,
\end{aligned}
$$

where $\omega_{\max} = \max_{(u,i) \in \mathcal{D}} |p_{u,i}(1 - p_{u,i})(e_{u,i}^{(1)} - e_{u,i}^{(0)})^2 / \hat{p}_{u,i}^2|$. The second equal comes from the proof (1) of the Corollary 3.

(2)

Similar to (1), we get

$$
\begin{aligned}
\text{Var}[\mathcal{E}_{\text{DR}}] &\leq \omega_{\max} \cdot \frac{1}{|\mathcal{D}|^2} \sum_{u,i \in \mathcal{D}} |P(r_{u,i} = 1) - \tilde{r}_{u,i}|^2 \\
&\leq \omega_{\max} \cdot \frac{1}{|\mathcal{D}|} \max_{u,i \in \mathcal{D}} |P(r_{u,i} = 1) - \tilde{r}_{u,i}|^2 \\
&= \omega_{\max} \cdot \frac{1}{|\mathcal{D}|} \cdot \left( \max_{u,i \in \mathcal{D}} |P(r_{u,i} = 1) - \tilde{r}_{u,i}| \right)^2 \\
&= \frac{\omega_{\max}}{|\mathcal{D}|} \cdot \big(\text{MCE}(g_\phi)\big)^2
\end{aligned}
$$

□