# OpenReview forum: "Doubly Calibrated Estimator for Recommendation on Data Missing Not At Random"
_ACM.org/TheWebConf/2024/Conference — TheWebConf24 Oral_

### Official Review · Reviewer_Bx7M · 2023-10-29

**Novelty:** 5
**Technical Quality:** 6

**Review:**

**Strengths**

1. **Interpretable theoretical analysis.** The analysis of the upper bound of the errors with expected calibration error (ECE) is quite understandable and well-connected to the discussion of calibration.

2. **Extensive experiments and ablation study.** Showing the correlation between ECE, AUC, and MSE supports the discussion of this paper well and demonstrates the benefits of the proposed approach. In addition, the ablation with "Only_ImpCal" and "Only_PropCal" shows why the "doubly" calibration is important. Ablation study with varying values of $K$ (number of user groups) also reports reasonable results. Overall, the empirical analysis in extensive and informative.

3. **Well-written paper.** The manuscript is easy to follow, and the claims are well-supported.


**Weaknesses**

1. **Related work on calibration should be discussed.** Although this is not an existing work on matrix factorization with missing not at random (MNAR), the calibration of the propensity is discussed in off-policy evaluation literature **(Raghu et al., 18)**, which seems relevant to this paper. Having a discussion of related work on calibration would be valuable for readers.

**(Raghu et al., 18)** Aniruddh Raghu, Omer Gottesman, Yao Liu, Matthieu Komorowski, Aldo Faisal, Finale Doshi-Velez, Emma Brunskill. "Behaviour Policy Estimation in Off-Policy Policy Evaluation: Calibration Matters." 2018.

**Questions:**

One thing that was unclear to me was the role of maximum calibration error (MCE) in this paper. From the definition, MCE should always be greater than ECE. Thus, when the upper bound is given by ECE, the upper bound based on MCE seems redundant. Are there any reasons to include MCE in this paper?

**Reviewer Confidence:**

3: The reviewer is confident but not certain that the evaluation is correct

**Scope:**

4: The work is relevant to the Web and to the track, and is of broad interest to the community

---

### Official Review · Reviewer_PeCP · 2023-11-02

**Novelty:** 4
**Technical Quality:** 5

**Review:**

**Summary**


This paper addresses the selection bias issue in recommender systems, where users typically rate their preferred items more frequently, leading to data that doesn't accurately represent the target population. The authors identify shortcomings in existing doubly robust (DR) estimators, which, despite their advanced bias correction for missing not at random (MNAR) data, rely on models that often produce miscalibrated imputed errors and propensity scores. To overcome these limitations, the authors introduce a "Doubly Calibrated Estimator" methodology. This new approach recalibrates both the imputation and propensity models by incorporating calibration experts for different user groups, enhancing the accuracy and reliability of the estimators. The proposed system, which is an orthogonal enhancement to existing DR estimators, demonstrates reduced bias and variance without requiring additional unbiased data, as confirmed through experiments on real-world datasets.


**Strengths**

- The paper tackles the practically important problem of recommendations from MNAR feedback


- The paper is well written and the arguments are easy to understand. In particular, the paper succeeds in motivating the proposed calibration methods well with brief theoretical analysis in the early part of the main text


- The paper's findings are tested through extensive experiments on both real-world and semi-synthetic datasets. Experiment design and choice of baselines look sufficient, and the provided results are promising and show the advantages of calibration in DR



**Weaknesses**


- Even though debiasing in recsys is a practically relevant and important area of research, it is extremely dense, so proposing a simple extension of DR might be considered a marginal contribution


- Even though the paper provides some theoretical analysis to show the connection between the bias&variance and (mis)calibration, I was not sure how tight the provided bounds are. Authors should be able to test the plausibility of the bounds using some synthetic data where we can calculate the true bias, variance, and calibration of methods.


- The paper does cover the well-established real-world datasets in this field, but all of them are extremely small. In real practice, we often face a system with million users and items, so in order to rigorously evaluate the methods’ real-world advantage and scalability, it would be better to apply the method to much larger datasets too


- Even though the paper performs offline experiments on well-known real-world datasets, it fails to perform online A/B test; so it might be risky to be overly optimistic about the empirical results of the paper until the proposed approach is verified in an online environment and provides some tangible business benefits

**Questions:**

- How novel is it to consider adding the calibration feature to DR, which is well known?


- How tight are the provided upper bounds of the bias and variance?


- How effective and scalable would the proposed method be in a system of practical size?

**Reviewer Confidence:**

4: The reviewer is certain that the evaluation is correct and very familiar with the relevant literature

**Scope:**

4: The work is relevant to the Web and to the track, and is of broad interest to the community

---

### Official Review · Reviewer_7Uqs · 2023-11-23

**Novelty:** 4
**Technical Quality:** 5

**Review:**

This paper investigated the calibration problem in the estimation of imputed errors and propensity in doubly calibrated estimators. It was proved that the bias of the DR estimator is controlled by the calibration error of propensity model, and the variance of the DR estimator is controlled by the calibration error of both propensity and imputed errors. Then the authors proposed a mixture-of-expert based model to enhance the calibration. It was proved that the proposed model have a better calibration, leading to better performance in three recommendation datasets.

**Questions:**

I was curious about the relationship between the bias of DR estimator and the calibration error of propensity model. It seems that this is the missing part of the theory. Can you prove that the bias of DR estimator is upper bounded by the calibration error of propensity model? If not, is there a counter-example, or is there any possibility to add some mild assumptions to make this work. I expect more discussions on this.

**Reviewer Confidence:**

3: The reviewer is confident but not certain that the evaluation is correct

**Scope:**

4: The work is relevant to the Web and to the track, and is of broad interest to the community

---

### Official Review · Reviewer_VGdR · 2023-11-30

**Novelty:** 6
**Technical Quality:** 6

**Review:**

**Summary:** The authors propose Doubly Calibrated Estimator that involves the calibration of both the imputation and propensity models, in which imputed errors and propensity scores are calibrated with calibration experts by assigning users to each expert through the assignment network.


**Strengths:**

- I am also working on this research topic, so I understand this is an important research. The proposed method is well-motivated and technically sound.

- The presentation of the proposed method is clear and easy to understand.

- Experimental results demonstrate the superiority of the proposed method over existing methods.

**Weaknesses:**

- My major concern is the missing related work, I encourage the authors to add the following references in their final version (although some of them are recently published) and discuss the relations and differences.

  - Some recent works [1-3] focusing on the propensity and imputation calibrations.

  - The relevant and state-of-the-art doubly robust methods [4-8].

  - Other relevant works for debiased recommendations [9, 10].

  - Please kindly note that [5] and [6] in the submitted manuscript are the same. Should remove one of them.

- (Minor) The presentation needs to be clearer.
  - As the main motivation for this paper, the authors claim "both logistic regression and neural networks have a tendency to generate overly confident predictions" in the Introduction Section, however, as it is rarely mentioned in debiased recommendations, it is not clear here what is the meaning of “overly confident predictions” for the imputed errors and learned propensities. I suspect that the authors are trying to say that the imputed errors and learned propensities are too small or too large, but I would encourage the authors to clarify this in the Introduction Section.
  - I also encourage the authors to conduct the experiment same as Fig. 1 on more/larger-scale dataset, such as KuaiRec.
  - It is unclear what do $\mu_1$ and $\mu_0$ means in Section 4.1.
  - It is unclear how to learn $c^{prop}(p)$ and $c^{imp}(\tilde r)$, from user embeddings or the pre-trained $\hat p$ and $\tilde r$?
  - In Section 4.2, the authors devise an assignment network to assign each user to an expert, by mapping the user embedding to the assignment probability. Why only consider user-specific calibration, instead of item-specific calibration, or user-item-specific calibration? Because the target population is defined as all user-item pairs, not users.
- (Minor) There are some incorrect equations.
  - The equation $c^{imp}(p)$ should be $c^{imp}(\tilde r)$ throughout Section 4.1.
  - The equation $1-\frac{r_{u, i}}{\bar p_{u, i}}$ in Eq.(14) should be $\frac{1-r_{u, i}}{\bar p_{u, i}}$,as a reweighted cross-entropy loss.
  - The punctuation before "where" in Corollary 3 should be a comma.

***

**References**

[1] Propensity matters: measuring and enhancing balancing for recommendation. In ICML 23.

[2] Removing hidden confounding in recommendation: a unified multi-task learning approach. In NeurIPS 23.

[3] Rating distribution calibration for selection bias mitigation in recommendations. In WWW 22.

[4] Doubly robust estimator for ranking metrics with post-click conversions. In RecSys 20.

[5] Reaching the end of unbiasedness: uncovering implicit limitations of click-based learning to rank. In SIGIR 22.

[6] Doubly robust estimation for correcting position bias in click feedback for unbiased learning to rank. In ACM Transactions on Information Systems 23.

[7] Who should be given incentives? Counterfactual optimal treatment regimes learning for recommendation. In KDD 23.

[8] CDR: Conservative doubly robust learning for debiased recommendation. In CIKM 23.

[9] Mitigating confounding bias in recommendation via information bottleneck. In RecSys 21.

[10] Towards resolving propensity contradiction in offline recommender learning. In IJCAI 22.

**Questions:**

- As previously stated, adding data analysis to support the assumptions would enhance the credibility and validity of the paper.

- Also, I recommend including a clear description of how to obtain the propensity $c^{prop}(p)$ and imputation $c^{imp}(\tilde r)$.

**Reviewer Confidence:**

4: The reviewer is certain that the evaluation is correct and very familiar with the relevant literature

**Scope:**

4: The work is relevant to the Web and to the track, and is of broad interest to the community

---

### Official Review · Reviewer_A9Ww · 2023-11-30

**Novelty:** 5
**Technical Quality:** 5

**Review:**

This paper proposes a doubly calibrated estimator that involves the calibration of both the imputation and propensity models and uses several calibration experts to help to learn a more accurate prediction model by a tri-level joint learning framework. Real-world experiment results demonstrate the effectiveness of the proposed calibration method.

**Pros:**

1: The research question of this paper is interesting and worth studying.

2: The proposed method that calibrates both the propensity model and imputation model is innovative.

3: This paper provides empirical studies to further enhance the motivation.

4: Real-world experiment results demonstrate the effectiveness of the proposed calibration method.

**Cons:**

1: How the proof of Theorem 2 holds? Why $\max_{\hat{p} \in [0,1]} |P(o = 1|h_\psi (x) = \hat p) − \hat p| = \max_{(u, i)\in D} | P(o_{u, i} = 1) − \hat{p}_{u, i} |$?

Specifically, when $\hat{p}$ is the same for all the user-item pairs, the $\max_{\hat{p} \in [0,1]}(\cdot)$ is not equivalent to $\max_{(u, i)\in D}(\cdot)$. For example, if we have all 100 user-item pairs with propensity scores $\hat p_{u, i}= 0.2$ and 20 pairs are with $o_{u, i}$. That is, the learned propensities have been already well-calibrated. Meanwhile, suppose half of them have $p_{u, i}=0.1$ and another half of them have $p_{u, i}=0.3$. Then $\max_{\hat{p} \in [0,1]} |P(o = 1|h_\psi (x) = \hat p) − \hat p| = 0 $ but $ \max_{(u, i)\in D} | P(o_{u, i} = 1) − \hat{p}_{u, i} |$=0.1.

2: The paper claims that the $c^{imp}_{\omega_k}$ in Eq. (14) relies on the embedding of the imputation model.

What is the embedding of a model? If the embedding means $x_{u, i}$, then the  $c^{imp}_{\omega_k}$ is independent of the imputation model, thus it should not be involved in the joint learning process of Algorithm 1.

Because neither the parameters of the imputation model $g_{\phi}$, nor the parameters of the prediction model $f_{\theta}$ affect the training process of $c^{imp}_{\omega_k}$.

3: The proposed method tries to control the upper bound. However, the upper bound is too loose by taking the $\max$ operation. How to prove that Eq. (15) can help calibration and help reduce ECE and MCE? Meanwhile, it is hard to guarantee that controlling the upper bound can effectively reduce the variance and the bias of the estimator.

4: What is the benefits of using $\beta_{u, k}$ compared with using $\alpha_{u, k}$? I think both of them are designed for estimating the assignment probability.

5: Why TDR-CL has lower running time and parameter numbers than DR-JL on the Yahoo!R3 dataset?

6: Lack of references. Since this work is highly correlated to the debiased recommendation and causal learning, the references [1-3] should be included and discussed.

[1] Jiawei Chen et al. 2022. Bias and Debias in Recommender System: A Survey and Future Directions. In ACM Transactions on Information Systems.

[2] Huishi Luo et al. 2023. A Survey on Causal Inference for Recommendation. In arXiv.

[3] Peng Wu et al. 2022. On the Opportunity of Causal Learning in Recommendation Systems: Foundation, Estimation, Prediction and Challenges. In IJCAI.

**Questions:**

Please look at the **Cons** for the questions.

**Reviewer Confidence:**

4: The reviewer is certain that the evaluation is correct and very familiar with the relevant literature

**Scope:**

3: The work is somewhat relevant to the Web and to the track, and is of narrow interest to a sub-community

---

### Decision · Program_Chairs · 2024-01-22

**Decision:**

Accept (Oral)

**Comment:**

The paper deals with recommendations where training data and the target distribution do not match. In particular, the setting is training data that is "missing not at random". The paper first argues that the common approach of doubly robust estimators has flaws as the estimators are not calibrated. This is demonstrated empirically. Then the paper suggests to model K "calibration experts" and to assign each user to one expert. This assignment and the experts are jointly learned with the imputation and prediction model. The proposed approach is evaluated and shows promising results.

 The reviews see the targeted problem as important. The method of calibrating both the imputation and propensity model is innovative. Most reviewers were domain experts and asked detailed technical questions during the rebuttal which the authors could answer to the satisfaction of the reviewers. This is an indicator for the technical soundness of the work.
 The paper is found to be well presented. The experiments are overall sound but one reviewer mentioned that the studied datasets are small and there are no online A/B experiments.

 Overall, this is an interesting paper that provides a technically sound solution to a relevant problem. The reviews mention several minor presentation issues that should be clarified in a camera ready version.